# WolBanking77: Wolof Banking Speech Intent Classification Dataset

**Abdou Karim Kandji**[1][*]    **Frédéric Precioso**[2]    **Cheikh Ba**[1]    **Samba Ndiaye**[3]

**Augustin Ndione**[3]

[1]University of Gaston Berger (UGB), Saint-Louis, Senegal
[2]Inria, Université Côte d'Azur (UniCA), Maasai team, Nice, France
[3]Cheikh Anta Diop University, Dakar, Senegal
{kandji.abdou-karim1,cheikh2.ba}@ugb.edu.sn
frederic.precioso@univ-cotedazur.fr
{samba.ndiaye,augustin.ndione}@ucad.edu.sn

## Abstract

Intent classification models have made a significant progress in recent years. However, previous studies primarily focus on high-resource language datasets, which results in a gap for low-resource languages and for regions with high rates of illiteracy, where languages are more spoken than read or written. This is the case in Senegal, for example, where Wolof is spoken by around 90% of the population, while the national illiteracy rate remains at of 42%. Wolof is actually spoken by more than 10 million people in West African region. To address these limitations, we introduce the Wolof Banking Speech Intent Classification Dataset (WolBanking77), for academic research in intent classification. WolBanking77 currently contains 9,791 text sentences in the banking domain and more than 4 hours of spoken sentences. Experiments on various baselines are conducted in this work, including text and voice state-of-the-art models. The results are very promising on this current dataset. In addition, this paper presents an in-depth examination of the dataset's contents. We report baseline F1-scores and word error rates metrics respectively on NLP and ASR models trained on WolBanking77 dataset and also comparisons between models. Dataset and code available at: wolbanking77.

## 1 Introduction

Wolof is spoken in Senegal, Gambia and Mauritania by more than 10 million people. Most of the population in Senegal speaks Wolof (around 90%) [2] which is essentially a language of communication between several ethnic groups. More details on Wolof are presented in the appendix section D.

The lack of digital resources for Wolof motivated us to build the WolBanking77 dataset. Wolof, being an oral language, it is essential to establish voice assistants that allow the population access to digital services and help address challenges such as enhancing financial inclusion and access to digital public services. The trade sector, for instance, which plays an important role in the economy of African countries, is predominantly driven by the informal economy [Ouattara et al., 2025]. We can also cite the agriculture, livestock, fishing, and transport sectors, which are all related to commercial activity and generate more employment compared to the formal sector [Martínez and Short, 2022]. Recent

---

[*]Corresponding author
[2]https://www.axl.cefan.ulaval.ca/afrique/senegal.htm

advances in artificial intelligence provide an good opportunity for these populations to benefit from digital technology. They can gain better control over financial transactions and minimize the risk of fraud due to language barriers. In fact, the language barrier often compel business owners in the informal sector to rely on a third party to consult their account balances or make payments on their behalf. This potentially exposes them to the risk of fraud [Anthony et al., 2024, Ouattara et al., 2025].

However, to build such a virtual assistant, it is necessary to understand human requests in order to provide appropriate responses. In Natural Language Processing (NLP), the field that deals with the understanding of human language, is Natural Language Understanding (NLU). In our case, we focus on both text and speech given that our objective is to work with African languages rooted in oral traditions. Before doing an NLU task, the first step is to understand the information contained in speech using Spoken Language Understanding (SLU). According to the World Bank Bank [2022], the rate of adult illiterate (see Appendix A.4) population in Senegal is 42%, which means that to facilitate access to a virtual assistant for these populations, it is essential to offer voice processing solutions. To determine the intent of a user request, Coucke et al. [2018] frame the problem as an Intent Detection (ID) classification task, performed either directly from text or from audio transcriptions.

Models based on neural networks has emerged in recent years in the field of ID [Gerz et al., 2021, Krishnan et al., 2021, Si et al., 2023b]. However, very few datasets in African languages Alexis et al. [2022], Mastel et al. [2023], Moghe et al. [2023], Mwongela et al. [2023], Kasule et al. [2024] have been explored. Even less in the field of e-banking for Sub-Saharan languages. Most existing datasets are in English Coucke et al. [2018], as their development requires significant financial and human resources. In this paper, we present an intent classification dataset in Wolof, an African and Sub-Saharan language, based on the Banking77 dataset Casanueva et al. [2020] which originally consists of 13,083 customer service queries labeled with 77 intents. In addition, our work also draws on the MINDS-14 dataset Gerz et al. [2021], which contains 14 intents derived from a commercial e-banking system and includes an audio component.

## 2 Related work

### 2.1 Existing datasets

Several datasets, both monolingual and multilingual, for NLU conversational question-answering system, have been published in the last decade. For instance, The Chatbot Corpus dataset composed of questions and answers and also The StackExchange Corpus based on the StackExchange platform both available on GitHub [3] and evaluated by [Braun et al., 2017]. More recently, MINDS-14 Gerz et al. [2021] a multilingual dataset in the field of e-banking has been made publicly available. When considering multilingual intent classification benchmarking, we can mention for instance the MultiATIS++ dataset Xu et al. [2020], which pertains to the aviation field, and has been expanded upon the original English ATIS dataset Price [1990] to six additional languages. More recently, the XTREME-S benchmark Alexis et al. [2022] aims at evaluating several tasks such as speech recognition, translation, classification and retrieval. XTREME-S brings together several datasets covering 102 various languages, including 20 languages of Sub-Saharan Africa.

With regard to the Wolof language, datasets have been published in the literature, particularly in the field of NLP and Automatic Speech Recognition (ASR). For example, some datasets include Wolof texts such as afriqa dataset Ogundepo et al. [2023] which deals with the question answering (QA) task, masakhaner versions 1 and 2 Adelani et al. [2021, 2022b] which targets the Name Entity Recognition (NER) task, UD_Wolof-WTB [4] or even MasakhaPOS [5] which addresses the part-of-speech tagging (POS) task. In addition, several datasets for the machine translation (MT) task such as OPUS, [6] FLORES 200 team et al. [2022], NTREX-128 Federmann et al. [2022] and MAFAND-MT [Adelani et al., 2022a]. In the field of ASR, several datasets containing Wolof have also been published, including ALFFA Gauthier et al. [2016], fleurs Conneau et al. [2023] and more recently KALLAAMA Gauthier et al. [2024]. The cited datasets cover fairly general domains, such as news and religion. However, to date, there is only one text dataset dedicated to the banking sector

---

[3]https://github.com/sebischair/NLU-Evaluation-Corpora
[4]https://github.com/UniversalDependencies/UD_Wolof-WTB
[5]https://github.com/masakhane-io/lacuna_pos_ner
[6]https://opus.nlpl.eu/

named INJONGO Yu et al. [2025] (with other domains such as home, kitchen and dining, travel and utility). This dataset includes slot-filling and intent classification tasks for 16 African languages including Wolof.

The authors Casanueva et al. [2020] explored few-shot learning scenario in addition to introducing the Banking77 dataset, which contains 77 intents and 13,083 examples. ArBanking77 Jarrar et al. [2023] is the Arabic language version of the Banking77 dataset Casanueva et al. [2020] in which the authors conducted simulations in low-resource settings scenario by training their model on a subset of the dataset. Other recent contributions to low-resource languages have been made, the authors Mastel et al. [2023] used Google Cloud Translation API to translate the ATIS dataset Price [1990] in Kinyarwanda and Swahili which are languages spoken by approximately 100 million people in East Africa. Similarly, Moghe et al. [2023] introduced the MULTI3NLU++ dataset for several languages including Amharic. Kasule et al. [2024] proposed a voice command dataset containing 20 intents in Luganda, designed for deployment on IoT devices.

## 2.2 Summary of contributions

In this work, 9,791 customer service queries from the Banking77 dataset Casanueva et al. [2020] was translated to French and Wolof by a team of linguistic experts from the Centre de Linguistique Appliquée de Dakar (CLAD). Additionally, this work introduces another dataset based on MINDS-14 Gerz et al. [2021], in which each query is paired with audio recordings from multiple speakers with diverse accents and ages (see Appendix A.3). The dataset includes 10 intents represented with both text and audio examples. The open source tool Lig-Aikuma Blachon et al. [2016] was used to produce the voice recordings. Our contributions include:

- An audio dataset with 263 sentences covering 10 intents in the banking and transport domains, including diverse voices and accents, making it the first of its kind in the Wolof language at the time of writing.
- A text dataset with 9,791 sentences translated from English to French and Wolof covering 77 intents in banking domain.
- The results obtained with state-of-the-art models in various tasks such as Automatic Speech Recognition (ASR) and Intent Detection (ID). We also provide training and evaluation code to support the reproducibility of experimental benchmarks.
- A dataset documentation (datasheets) for WolBanking77.

All datasets and source codes are released under a CC BY 4.0 license to stimulate research in the field of NLP for low-resource African languages.

## 3 Data collection

In this section, we present the main information on the creation of WolBanking77, a more detailed description can be found in Appendix E.

### 3.1 Textual data

The text dataset contains a total of 9,791 sentences and 77 intents from the English train set of Banking77 Casanueva et al. [2020], manually translated to French and Wolof thanks to a team of linguistic experts from the Centre de Linguistique Appliquée de Dakar (CLAD). The Wolof version was translated and localized according to the local context, for instance, "*ATM*" and "*app*" translated as "*GAB*" and "*aplikaasiyoŋ*", see the example in figure 1.

Duplicated sentences translated in Wolof was removed for the ID task to avoid many-to-one translations from English to Wolof. Two versions of the dataset are reported on table 2. The first version is comprised of 5k samples which is a sub-sample of the second version of 9,791 samples. Train set is 80% of both versions and test set represents 20%. Note that intents are unbalanced, the most represented intent has a frequency of 200 while the least represented one has a frequency of 24.

Some statistics were collected on the data translated into French and Wolof. Statistics on table 1 show a slightly higher number of words per query for French (83) compared to Wolof (81).

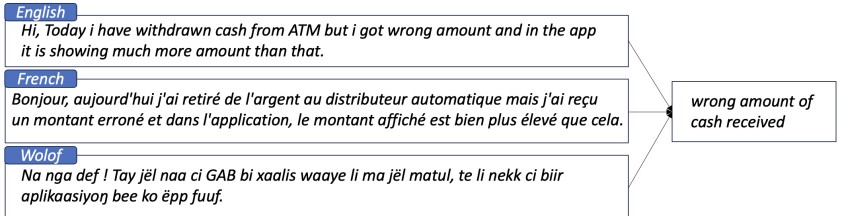

Figure 1: A query translated in French and Wolof

Table 1: WolBanking77 dataset statistics

|  | WOLOF | FRENCH |
|---|---|---|
| Min Word Count | 2 | 2 |
| Max Word Count | 81 | 83 |
| Mean Word Count | 12.22 | 12.47 |
| Median Word Count | 10 | 10 |

Table 2: WolBanking77 dataset split

| SPLIT | 5k sample | all |
|---|---|---|
| TRAIN SET | 4,000 | 7,832 |
| TEST SET | 1,000 | 1,959 |

## 3.2 Audio recordings and transcriptions

The audio corpus based on MINDS-14 Gerz et al. [2021] consists of 186 queries and 77 responses, initially translated from French into English by a team of linguistic experts from the Centre de Linguistique Appliquée de Dakar (CLAD). The dataset also includes a phonetic transcription of the Wolof text to facilitate the correct pronunciation of sentences by different speakers. Additional intents was added to the initial MINDS-14 dataset, namely: *OPEN_ACCOUNT*, for information related to opening a bank account, *BUS_RESERVATION*, for booking a transport bus (see Appendix A.5), *TECHNICAL_VISIT* for scheduling a vehicle inspection appointment. *TRANSFER_MONEY* for the intent to transfer money and *AMOUNT* to specify the amount to be transferred. See table 3 for the complete list of intents in the audio dataset.

Figure 2 shows top 5 most frequent words in the datastet after excluding stopwords. The dataset contains 272 unique words and 3,204 audio clips. The audio clips are in WAV format, single channel, with a sampling rate of 16 kHz. Sentences have an average duration of 4,815 ms. The intents *'CASH_DEPOSIT'*, *'FREEZE'*, *'LATEST_TRANSACTIONS'* contain the longest queries, whereas *'AMOUNT'*, *'BUS_RESERVATION'*, *'OPEN_ACCOUNT'* and *'TRANSFER_MONEY'* generally have shorter query durations. The total duration of the dataset is approximately 4 hours and 17 minutes.

Table 3: List of intents.

| intent | domain |
|---|---|
| BALANCE | e-banking |
| CASH_DEPOSIT | |
| FREEZE | |
| LATEST_TRANSACTIONS | |
| PAY_BILL | |
| OPEN_ACCOUNT | |
| TRANSFER_MONEY | |
| AMOUNT | |
| BUS_RESERVATION | transport |
| TECHNICAL_VISIT | |

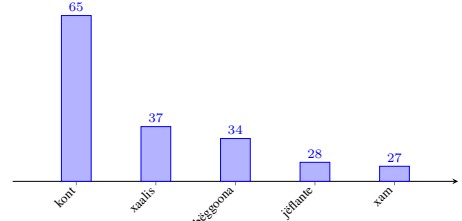

Figure 2: Top 5 most frequent words: kont (account), xaalis (money), bëggoona (I wanted to), jëflante (operation), xam (know).

The dataset was collected with the participation of students from Cheikh Anta Diop University in Dakar (UCAD), specifically from the Faculty of Letters and Human Sciences. Each participant recorded their voice using the elicitation mode of the Lig-Aikuma software [Blachon et al., 2016]. A total of 186 utterances were recorded by participants in a controlled environment. The elicitation mode of the Lig-Aikuma software was installed on an Android tablet. At the start of each session, information about the participant are requested, including native language, region of origin (see figure 3), gender, year of birth and name. This information is subsequently stored as metadata on the tablet's

memory card. Note that, the scores presented in figure 3 are not representative of the population of each region. We tried to cover various ethnic groups in the country in order to have diverse accents and dialects as described in Appendix A.2. Additional details on demographic data and distribution can be found in Appendix A.1.

To anonymize participant names, the system generates a user_id to replace the actual names (further details on the ethical collection and processing of personal data are provided in Appendices A.11 and C). Next, a text file containing the sentences to be pronounced is selected. During recording, sentences were presented one at a time, with the option to cancel if a mispronunciation occurred at any stage. In terms of gender diversity, a total of 31 recording sessions were conducted, including 14 males, 14 females and 3 unspecified. The text and audio data were split with 80% allocated for train set and 20% for test set. Prior to data splitting, preprocessing steps were applied, including the removal of punctuation marks and numbers, and conversion of text to lowercase. Corrupted audio files were also removed from the dataset.

## 4    Tasks & Settings

### 4.1    Automatic Speech Recognition

Automatic Speech Recognition (ASR) refers to the technology that enables a model to recognize and convert spoken language into text. ASR systems are widely used in various applications such as voice assistants, transcription services and customer service automation. Significant research has been conducted in this area, including Listen Attend and Spell by Chan et al. [2016], an end-to-end speech recognition system based on a sequence-to-sequence neural network. Graves et al. [2006] proposed Connectionist Temporal Classification (CTC) model used for training deep neural networks in speech recognition as well as other sequential problems where there is no explicit alignment information between the input and output. More recently, the NVIDIA team Żelasko et al. [2025] developed the Canary Flash model, a variant of Canary models Puvvada et al. [2024] notable for being both multilingual and multitask. This model achieved state-of-the-art results on several benchmarks, including ASR.

Another state-of-the-art model developed by Microsoft and named Phi-4-multimodal-instruct Microsoft et al. [2025] has recently been released to address ASR tasks, as well as vision and text applications. Phi-4-multimodal achieves an average WER score of 6.14% and currently ranks second on Huggingface's OpenASR leaderboard [7]. To evaluate ASR models, Word Error Rate (WER) scores are reported in Section 5.4.

---

[7]https://huggingface.co/spaces/hf-audio/open_asr_leaderboard

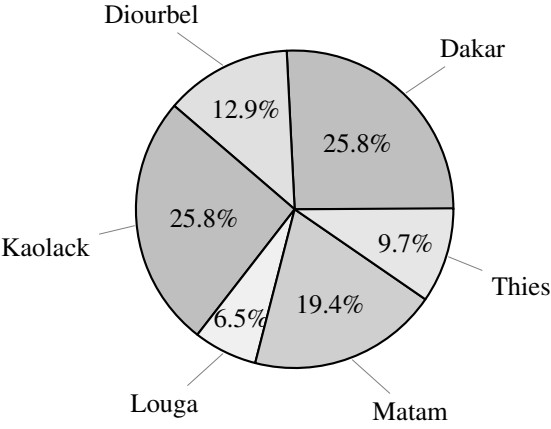

Figure 3: Senegal region repartition by speaker

## 4.2 Intent Detection

Intent Detection (ID) is an NLP task that involves classifying a sentence to identify its underlying intent. Several contributions have been made in the field. For instance, Gerz et al. [2021] published MINDS-14, the first training and evaluation dataset for the ID task using spoken data. It includes 14 intents derived from e-banking domain, with spoken examples in 14 different languages. Si et al. [2023a] introduced SpokenWOZ, a large-scale speech-text dataset for spoken Task-Oriented Dialogue in 8 domains and 249 hours of audio. Casanueva et al. [2020] published BANKING77, a dataset comprising 13,083 examples across 77 intents in banking domain.

In this article, we aim to assess the challenging potential of our dataset to better highlight its relevance to the community by evaluating downstream tasks using progressively sophisticated ML Workflows. We first consider classic machine learning algorithms such as k-nearest neighbor (KNN), support vector machine (SVM), linear regression (LR) and naive bayes (NB) as initial baselines. The results are reported in Appendix A.9. We then consider slightly more sophisticated approaches, using the LASER sentence encoder Heffernan et al. [2022] pre-trained on several languages, including Wolof, for sentence encoding, followed by standard classification models such as Multi-Layer Perception (MLP) and Convolution Neural Networks (CNN). Results are reported in Appendix A.10. Subsequently, more advanced models based on BERT are evaluated in zero-shot-classification and few-shot-classification mode, and the same models are fine-tuned on the WolBanking77 dataset for comparison. Finally, several benchmarks of recent Small Language Models (SLM) such as **Llama-3.2-3B-Instruct** [8] and **Llama-3.2-1B-Instruct** [9] are presented. To evaluate ID models, *F1*, *precision* and *recall* metrics are reported in Section 5.4.

# 5 Dataset evaluation & Experiments

In this section, pre-trained models in multiple languages, including African languages, are presented for ID and ASR tasks. Details of the hyperparameter settings are provided in Appendix A.6. The results demonstrate the value of the WolBanking77 dataset for the community, highlighting its potential to improve ID task in Wolof and to serve as a basis for extending approaches to other low-resource languages.

## 5.1 Intent detection models

The next sections present and detail the pre-trained models used in the ID task. Experiments for zero and few shot classification are also discussed.

### 5.1.1 Zero-shot classification with WolBanking77

Zero-shot text classification is a NLP task in which a model, trained on labeled data from specific classes, can classify text from entirely new, unseen classes without additional training. In this experiments WolBanking77 provides the new, unseen dataset. To simulate Zero-shot text classification (more details in Appendix A.7), following pre-trained models are considered:

**BERT base (uncased)**: A transformers model pre-trained on English introduced by Devlin et al. [2019], using a Masked Language Modeling (MLM) objective.

**Afro-xlmr-large**: Based on XLM-R-large Conneau et al. [2020] using MLM objective, this model was published by Alabi et al. [2022] and pre-trained on 17 African languages but not Wolof, while still demonstrating in the original paper good scores for NER task on Wolof.

**AfroLM_active_learning**: Based on XLM-RoBERTa (XLM-R) using MLM objective Ogueji et al. [2021], this model was published by [Dossou et al., 2022]. AfroLM_active_learning is a Self-Active Learning-based Multilingual Pretrained Language Model for 23 African Languages including Wolof.

**mDeBERTa-v3-base-mnli-xnli**: Based on DeBERTaV3-base He et al. [2023, 2021], this model was published by Moritz et al. [2022] and pre-trained on the CC100 multilingual dataset Conneau et al. [2020], Wenzek et al. [2020] with 100 different languages including Wolof. This model can perform Natural Language Inference (NLI) on 100 languages.

---

[8]https://huggingface.co/meta-llama/Llama-3.2-3B-Instruct
[9]https://huggingface.co/meta-llama/Llama-3.2-1B-Instruct

**AfriTeVa V2 Base**: A multilingual T5 model released by Oladipo et al. [2023] and pre-trained on WURA dataset Oladipo et al. [2023] covering 16 African Languages not including Wolof.

F1-score metrics for Zero-shot text classification are reported in Section 5.4.

### 5.1.2 Few-shot classification with WolBanking77

Given the results obtained in zero-shot classification, it is relevant to leverage manually annotated data to improve the generalization capacity of existing models on the dataset presented in this article, WolBanking77. All pre-trained models cited in section 5.1.1 are selected for the few-shot classification to measure the gap between large pre-trained models with and without a small number of Wolof samples from WolBanking77 used for fine-tuning. F1-score metrics for few-shot text classification are reported in Section 5.4.

### 5.2 ASR System

State-of-the-art ASR are selected as baseline models, pre-trained on multiple languages with the possibility of fine-tuning them for a specific language and domain. A description of these models, with additional details, is provided below:

**Canary Flash**: Canary Flash Żelasko et al. [2025] has been pre-trained on several languages (English, German, French, Spanish) and on various tasks such as ASR and translation. Canary Flash is based on Canary Puvvada et al. [2024] that is an encoder-decoder model whose encoder is based on the FastConformer model Rekesh et al. [2023] and the decoder on the Transformer architecture [Vaswani et al., 2017]. Canary has the distinctive feature of concatenating tokenizers Dhawan et al. [2023] from different languages using SentencePiece.[10] In this work, English, Spanish, French and Wolof languages are concatenated with Canary's Tokenizer. These tokens are then transformed into token embedding before being fed into the Transformer decoder. In addition to the tokens of each language, Canary uses 1,152 special tokens representation. At the time of writing, Canary has three variants: canary-1b, canary-1b-flash, and canary-180m-flash. Canary-1b-flash version is chosen for the experiments because of its multilingual support. The canary-1b-flash model was trained on 85K hours of speech data, including 31K hours of public data (FLEURS Conneau et al. [2023], CoVOST v2 Wang et al. [2021b]) and the remainder on private data. The Mozilla CommonVoice 12 dataset Ardila et al. [2020] was used as validation data for each language.

**Phi-4-multimodal-instruct**: Phi-4-multimodal is a multimodal Small Language Model (SLM) supporting image, text and audio within a single model. It can handle multiple modalities without interference thanks to the Mixture of LoRAs [Hu et al., 2022]. To enable multilingual inputs and outputs, the tiktoken tokenizer [11] is used with a vocabulary size of approximately 200K tokens. The model is based on a Transformer decoder Vaswani et al. [2017] and supports a context length of 128K based on LongRopE [Ding et al., 2024]. For the speech/audio modality, several modules have been introduced, including an audio encoder composed of 3 CNN layers and 24 Conformer blocks [Gulati et al., 2020]. To map 1024-dimensional audio features to the 3072-dimensional text embedding space, 2 MLP layers are used in the Audio Projector module. Note that $LoRA_A$ was used to all attention and MLP layers with a rank of 320. Phi-4-multimodal was trained on 2M hours of private speech-text pairs in 8 languages. A second post-training phase using Supervised Fine Tuning (SFT) on speech/audio data pairs was then conducted. For the ASR task, this included 20k hours of private data and 20k hours of public data across 8 languages. The SFT data follows the format below:

```
< |user| >< audio > {task prompt} < |end| >< |assistant| > {label} < |end| >
```

The task prompt designates the specific task on which the model is to be fine-tuned.

Phi-4-multimodal outperforms nvidia/canary-1b-flash Żelasko et al. [2025], WhisperV3 1.5B Radford et al. [2023], SeamlessM4T-V2 2.3B Communication et al. [2023], Qwen2-audio 8B Chu et al. [2024], Gemini-2.0-Flash Team et al. [2023] and GPT-4o OpenAI et al. [2024] on the OpenASR dataset leaderboard [12] with a WER score of 6.14.

---

[10] Google Sentencepiece Tokenizer

[11] Tiktoken Tokenizer

[12] https://huggingface.co/spaces/hf-audio/open_asr_leaderboard

**Distil-whisper-large-v3.5**: Distil-whisper-large-v3.5 is based on Whisper introduced by Radford et al. [2023], which was trained through supervised learning on 680,000 hours of labeled audio data. The authors demonstrated that models trained with this scale could generalize to any dataset through zero-shot learning, meaning they can adapt without requiring fine-tuning for specific datasets. The training data covered 97 different languages. Several versions of Whisper have been released in recent years, including version 3 on which the distil-whisper-large-v3.5 model was built by knowledge-distillation following the methodology described by Gandhi et al. [2023]. Distil-whisper-large-v3.5 was trained on 98k hours of diverse filtered datasets such as Common Voice Ardila et al. [2020], LibriSpeech Panayotov et al. [2015] , VoxPopuli Wang et al. [2021a] , TED-LIUM Hernandez et al. [2018], People's Speech Galvez et al. [2021], GigaSpeech Chen et al. [2021], AMI Carletta et al. [2006], and Yodas [Li et al., 2023].

## 5.3 Model training

All models presented in this article are based on *Pytorch* library [Paszke et al., 2019]. All experiments were conducted on Runpod [13] and Kaggle.[14] P100 GPU (16GB VRAM) was used to finetune BERT-Base and mDeBERTa-v3-base. RTX 4090 GPU for Afro-xlmr-large, AfroLM_active_learning, AfriteVa V2 and Llama-3.2 models. For Zero-shot and Few-shot classification, RTX 2000 Ada GPU (16 GB VRAM) was used. Few-shot classification was performed using *SetFit huggingface* library [Tunstall et al., 2022]. *2-shots* refers to 2 samples per intent while *8-shots* represents 8 samples per intent. All pre-trained text models are hosted on *huggingface platform*, they were fine-tuned using *transformers* library Wolf et al. [2020] except for Llama-3.2 which was fine-tuned using *torchtune* library [torchtune maintainers, 2024].

MPL and CNN models were trained from scratch for 200 epochs. The MLP model consists of a single 800-dimensional hidden layer, a ReLU activation layer, and a fully connected output layer. For CNN, a Conv1d layer followed by a ReLU activation layer and a fully connected layer for output classification. The Wolbanking77 data was structured in the form of a prompt, as illustrated in Appendix A.8, to enable the Llama-3.2 model to generate the correct intent.

Canary Flash and Phi-4-multimodal-instruct ASR models were fine-tuned on the Wolbanking77 audio dataset using the *NVIDIA NEMO* framework Kuchaiev et al. [2019] and *Huggingface Transformers* library, respectively, on an A100 SXM GPU (80GB VRAM). In contrast, Distil-whisper-large-v3.5 was fine-tuned using the *Huggingface Transformers* framework on an RTX 2000 Ada GPU (16 GB VRAM). All ASR models were fine-tuned for 1,000 steps.

## 5.4 Results and Discussions

Figure 4: Zero-shot, few-shot and fine-tuning (FT) results (in % for F1-score) of multilingual pre-trained models on WolBanking77.

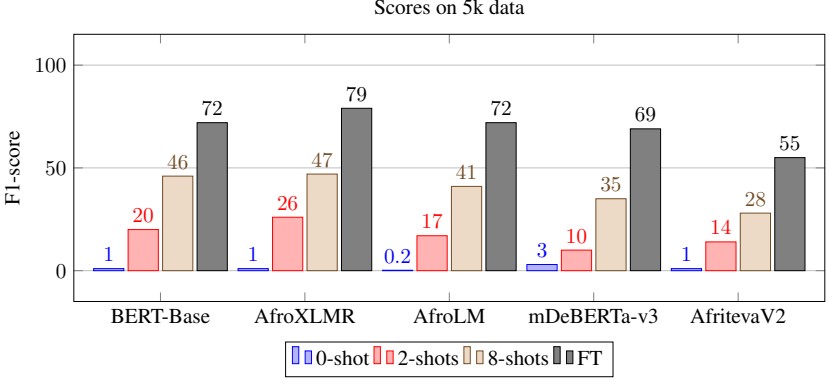

Figure 4 shows the weighted average F1-score results for zero-shot and few-shot classification. AfroXLMR model outperforms all multilingual pre-trained models in few-shot and fine-tuning

---
[13]https://www.runpod.io/
[14]https://www.kaggle.com/

settings (with an F1-scores of 79% on 5k samples), followed by BERT-Base and AfroLM. This results also show that existing models, even when pre-trained on Wolof, do not perform well and that our dataset presents specific challenges.

Figure 5: Zero-shot and fine-tuning (FT) results (in %) on multilingual pre-trained models.

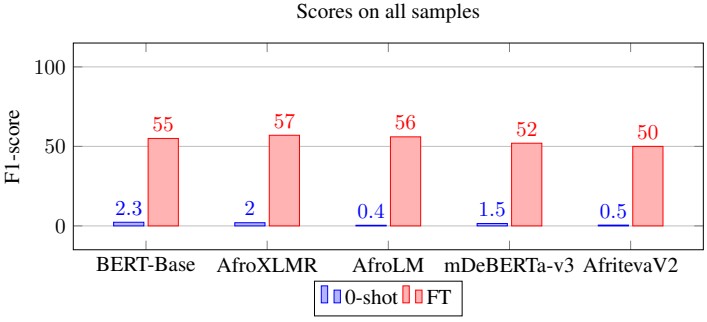

Experiments were conducted on the entire WolBanking77 dataset and reported in figure 5. Similar to the experiments performed on the 5k sub-sample of WolBanking77, results shows that AfroXLMR achieves slightly better scores compared to BERT-Base and AfroLM. These findings highlight that ID task for low-resource languages remains a challenging task, even for state-of-the-art small language models, as shown in Table 4. A comparison was also conducted for ASR models on the audio

Table 4: Precision, recall and f1-score scores for Small Language Models on WolBanking77. Best models are highlighted in lightgray.

| Model | 5k samples | | | All samples | | |
|---|---|---|---|---|---|---|
| | Precision | Recall | F1 | Precision | Recall | F1 |
| Llama-3.2-1B-Instruct | 0.74 | 0.71 | 0.72 | 0.52 | 0.45 | 0.46 |
| Llama-3.2-3B-Instruct | 0.76 | 0.75 | 0.75 | 0.56 | 0.55 | 0.55 |

part of WolBanking77 dataset. Special characters were removed from the text, and all text was converted to lowercase. We can observe from the results with 4 hours of speech data in Table 5 that, Canary-1b-flash with a WER score of 0.59% outperforms Phi-4-multimodal-instruct (WER 3.1%) and more particularly Distil-whisper-large-v3.5 (WER 4.63%). All pre-trained ASR models were fine-tuned on WolBanking77 audio dataset for 1000 steps. The results indicate that strong performance can be achieved with relatively little data, which is promising for WolBanking77 and other low-resource language contexts.

Table 5: Word Error Rates (WER) with 4 hours of speech. Training time is reported in minutes.

| Model | Parameters | Training time | Steps | WER |
|---|---|---|---|---|
| Phi-4-multimodal-instruct | 5.6B | 32 | 1000 | 3.1% |
| Distil-whisper-large-v3.5 | 756M | 44 | 1000 | 4.63% |
| **Canary-1b-flash** | 1B | 20 | 1000 | **0.59%** |

# 6    Long-term Support and Future Work

Long term support includes updates to the text and audio recordings, improvements to the simulation setup script, and potential bug fixes. Free access is provided to various versions of the dataset and source code under a CC BY 4.0 license to accelerate research in the field. It is planned to add more audio recordings in diverse environments to enhance model robustness. Text data corresponding to possible responses for each intent will be shared for potential use in Text-To-Speech applications. Additionally, the dataset can be annotated for slot-filling tasks.

# 7 Conclusion

Access to financial services or public transportation services can be facilitated for illiterate people by providing them with voice interfaces in their language of communication. In this paper, an Intent Detection dataset is presented in Wolof language in two modalities which are voice and text. Additionally, dataset description and benchmarks are presented. WolBanking77 is published and shared freely under CC BY 4.0 license along with the code and datasheets Gebru et al. [2021] with the aim of inspiring low budget research into low-resource languages.

## Acknowledgments and Disclosure of Funding

This work is supported by the Partnership for Skills in Applied Sciences, Engineering and Technology (PASET) - Regional Scholarship and Innovation Fund (RSIF) under Grant No.:B8501L10014. We also thank the entire CLAD team and in particular: Professor Souleymane Faye and all the translators who participated in this project.

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

# A    Technical Appendices and Supplementary Material

## A.1    WolBanking77 supplementary details

*Is the Wolof language as used in Mauritania and Gambia substantially distinct from the Senegalese variety, such that the demonstrated performance on the audio dataset may not generalise to these Wolof-speaking populations?*

The Wolof language used in Mauritania and Gambia is not substantially different. However, the differences are to be seen rather in linguistic evolutions and in particular with the borrowings from foreign languages present, in particular Arabic and English. These differences may pose a problem for the generalization of models trained on the audio data that do not yet cover these variants of Wolof. This is also present in the accent of the Wolof locutors which highly differs under the effect of the official language (English or Arabic). This has a strong impact on Wolof words pronunciation.

*It would also be useful to know the intersection of age and literacy in the population. If older speakers, for example, were proportionally less literate in the language, this might argue for strengthening the representation of older voices in the dataset, given that this development is aimed at improving accessibility through voice-interactive technology.*

According to statistics (from Agence Nationale de la Statistique et de la Démographie ANSD [2013]), the literacy rate is higher among the younger populations of 10-14 years old and 15-19 years old, with 58.1% and 64.1% respectively. However, the literacy rate decreases among older populations over 70 years of age (between 13% and 20%). In view of these figures, we intend to take the representation of the older population in future versions of the dataset.

*Is the digital infrastructure (mobile or desktop internet services) sufficient in all regions to support the deployment of the proposed future technology?*

The mobile phone and internet coverage is now very high in Senegal, the WhatsApp application for example is used throughout the country.

*Among the 77 text intents and 10 speech intents, is there any association between these intents? Has the possibility of multiple intents existing in one sample been considered?*

The text intent dataset and the speech intent dataset have been independently created. However, it could be possible to associate some speech intents with text intents such as BALANCE, CASH_DEPOSIT, FREEZE and TRANSFER_MONEY. It is indeed possible that a sample can belong to several intents, however this possibility has not been considered in this work.

## A.2    Linguistic variation in the Wolof language across Senegal

Wolof has several dialects, including Bawol, Kajoor, Jolof, and Jander; This dialectal diversity does not prevent inter-understanding. However, there is a lesser degree of inter-understanding between these varieties and that spoken by the Lebous (an ethnic group living mainly in the Cape Verde peninsula, i.e. in certain localities in the Dakar region).

## A.3    Speakers age distribution

The age distribution of the participants are presented in table 6.

Table 6: Age distribution of the speakers

| Stat | Age |
| --- | --- |
| mean | 26 |
| std | 4 |
| min | 22 |
| 25% | 23 |
| 50% | 25 |
| 75% | 27 |
| max | 36 |

## A.4 Illiteracy in Senegal

We use the word "illiterate" in the sense of not being able to understand the official language, which is French. " In Senegal, ANSD [2021] reports an overall illiteracy rate of 48,2%, reaching 62,7% in rural area. Literacy rate relates to the official language of a country. In Senegal, the official language is French but is seldom spoken by the population in their daily lives. Senegalese people primarily use their native languages or Wolof, as a vehicular language, to communicate." [Gauthier et al., 2024]

The school dropout rate is high in Senegal and this has resulted in a significant number of people who went to school and left without acquiring basic skills in the official language (French). As a result, in our situation, basic skills in French are lacking for more than half of the Senegalese population. One of the solutions has been to resort to literacy in national languages with satisfactory results with the experiments in Pulaar carried out by NGO TOSTAN and the examples of functional literacy carried out by UNESCO. Despite this, a large segment of the population still remains in a situation where basic linguistic skills, either in the official language or in one of the national languages, are acquired very little or not at all.

Literacy, considering the various languages present, but also the formal and informal systems, is progressing in Senegal even if it remains below 60% for adults according to statistics from UNESCO or the World Bank [macrotrends, 2022]. According to these same sources, we note that among young people aged 15 to 24, the rate is higher and is around 78%, a sign of this progression. It is clear that there is a disparity on two levels, firstly, at the gender level, in fact adult women are far more victims of illiteracy than men of the same age group and also, urban areas suffer less than rural areas (UNESCO) [countryeconomy, 2023].

## A.5 Transportation data

The main goal for creating this dataset is to design a speech chatbot for Wolof, in order to equip illiterate people from Senegal with an AI tool allowing them to access banking services, as well as buying a transit pass from public transportation, etc.

Adding transportation data, is thus not only a way to evaluate the model's ability to recognize out of context intents, but also to integrate complementary services into a mobile money application. Beyond making financial transactions with mobile money, it is possible to integrate other services such as transportation or bill payments (electricity, water, etc.). The fields are different but the kind of services (and so the sentences) we want to address are pretty similar in all of them.

## A.6 Hyperparameters

Table 7: Hyperparameters for NLP models. mDeBERTa-v3* and AfritevaV2 have same hyperparameters.

| Model | bert-base-uncased | AfroXLMR | mDeBERTa-v3* | AfroLM | Llama3.2 |
|---|---|---|---|---|---|
| Learning Rate | 2e-05 | 2e-05 | 2e-05 | 2e-05 | 3e-4 |
| Train Batch Size | 32 | 4 | 8 | 16 | 4 |
| Eval Batch Size | 32 | 8 | 8 | 8 | 4 |
| Warmup ratio | 0.1 | 0.1 | 0.1 | 0.1 | - |
| # epochs | 20 | 20 | 20 | 20 | 20 |

NLP models hyperparameters are reported on table 7, AdamW_torch_fused is the default optimizer for NLP models except for Llama3.2 that is AdamW.

Table 8: Hyperparameters for ASR models.

| Model | Canary-1b-flash | Distil-whisper-large-v3.5 | Phi-4 |
|---|---|---|---|
| Learning Rate | 3e-4 | 1e-5 | 1e-4 |
| Train Batch Size | - | 8 | 16 |
| Eval Batch Size | 8 | 8 | - |
| Gradient Accumulation Steps | - | 1 | 1 |
| Lr Scheduler Warmup Steps | 2500 | 500 | - |
| Optimizer | AdamW | AdamW | AdamW_torch |
| # steps | 1000 | 1000 | 1000 |

Table 8 shows the hyperparameters used to train Phi-4-multimodal-instruct, Distil-whisper-large-v3.5 and Canary-1b-flash. Hyperparameters has been chosen by following HuggingFace Wolf et al. [2020] and NVIDIA Nemo Kuchaiev et al. [2019] documentations. AdamW has been used as a default optimizer.

Table 9: Best hyperparameters for LASER3+CNN.

| Model | LASER3+CNN |
|---|---|
| Learning Rate | 0.013364046097018405 |
| Last size of non classification layer | 128 |
| Batch Size | 2 |

## A.7 Zero-shot classification

We consider the models listed in Section 5.1.1 for zero-shot classification. We use the pre-trained Masked Language Modeling (MLM) models (i.e. encoder models) on the datasets mentioned in this section.

We then use these models to embed each sentence and to project it onto the embedding of the intent classes for WolBanking dataset without additional training.

To run a model in zero-shot mode, we use the huggingface framework with a pipeline configured as "zero-shot-classification". This pipeline can be used to classify sequences into any of the specified class names. In this case, the pipeline uses the pre-trained model to perform inference and retrieve the logits at the model output. A softmax function is then applied to the logits to generate the probabilities of each class.

## A.8 Text prompt for Llama3.2

Following prompt format was used to train the Llama3.2 model.

```
Classify the text for one of the categories:

<text>
{text}
</text>

Choose from one of the category:
{classes}
Only choose one category, the most appropriate one. Reply only with the
↪  category.
```

{text} is a variable that can be one of the following sentences for example:

*Dama wara yónnee xaalis, ndax mën naa jëfandikoo sama kàrtu kredi?* (*I need to transfer some money, can I use my credit card?*)

*Jotuma xaalis bi ma ñu ma waroon a delloo.* (*I have not received a refund.*)

*Ci lan ngeeni jëfandikoo kàrt yi ñuy sànni?* (*What do you use disposable cards on?*)

*Ndax amna anam buma mëna toppe lii?* (*Was there a way for me to get tracking for that?*)

*Ban diir la wara am ngir yonnee xaalis Etats-Uni?* (*How long does it take for deliver to the US?*)

## A.9 Reference scores from classic Machine Learning Techniques

Machine learning models such as KNN, SVM, Logistic Regression (LR) and Naive Bayes (NB) with Bag-Of-Words are used as baselines as well as CNN and MPL with LASER3 as sentence encoder. Table 10 shows ML baselines comparison. Results shows that Bag of Words combined with Linear Regression outperforms the other ML models with an F1-score of 53%. However, LR and SVM achieve same results (F1-score of 68%) on 5k split.

Table 10: ML baselines models precision, recall and f1-score results on WolBanking77. Best models are highlighted in lightgray.

| Model | 5k samples | | | All samples | | |
| | Precision | Recall | F1 | Precision | Recall | F1 |
|---|---|---|---|---|---|---|
| BoW+KNN | 0.48 | 0.39 | 0.39 | 0.37 | 0.31 | 0.31 |
| BoW+SVM | 0.70 | 0.69 | 0.68 | 0.49 | 0.48 | 0.48 |
| BoW+LR | 0.70 | 0.69 | 0.68 | 0.54 | 0.53 | 0.53 |
| BoW+NB | 0.54 | 0.51 | 0.50 | 0.28 | 0.26 | 0.26 |

## A.10 Pretrained Sentence Encoder

**LASER3:** Heffernan et al. [2022] propose an improved version of LASER by training the model on multiple languages with the goal of encoding them within a shared representation space. Their method involves training a teacher-student model that combines supervised and self-supervised approaches with the aim of training the model on low-resource languages. This approach makes it possible to cover 50 African languages, including Wolof. Teacher-student training is employed to avoid training a new model from scratch each time a new language needs to be encoded. Some of the African languages originate from the Masakhane project [15] as well as the EMNLP'22 workshop.[16] The authors made some modifications to the LASER architecture such as replacing BPE with SPM (SentencePiece Model), an unsupervised text tokenizer and detokenizer that does not rely on language-specific pre or postprocessing, and adding an upsampling step for low-resource languages. The LASER sentence encoder trained on the public OPUS corpus [17] is used as the teacher model and renamed by **LASER2**, while the student model is referred to as **LASER3**. A student model is trained for each language covered in Masked Language Modeling objective, after which a multilingual distillation approach is applied by optimizing the cosine loss between the embeddings generated by the teacher and the student. The student architecture has been replaced by a 12-layer transformer instead of a 6-layer BiLSTM of the original LASER. For the Wolof language, the model was trained on 21k bitexts which are pairs of texts in two different languages that are translations of each other and 94k sentences using training distillation in addition to Masked Language Modeling. The experimental results show a clear improvement of LASER3 encoder for Wolof with a score of an xsim error rate of 6.03 (a margin-based similarity score by Artetxe and Schwenk [2019]) compared to the original LASER encoder which produced a score of 70.65.

The pre-trained LASER3 sentence encoder, which vectorizes each sentence and all possible combinations of classification models (MPL and CNN) are compared in the results reported in Table 11. Note that all classification models were trained from scratch. For all the evaluations, punctuation are removed and each sentence was converted to lowercase.

Table 11 presents the weighted average results of LASER3 combined with classification models (MLP, CNN). LASER+MLP achieves the best performance with an F1-score of 55% on 5k samples and 42% on the full dataset, compared to LASER+CNN which produced poor results with a low

---

[15]https://www.masakhane.io/

[16]https://www.statmt.org/wmt22/large-scale-multilingual-translation-task.html

[17]https://opus.nlpl.eu/

Table 11: LASER3 with classification models precision, recall and f1-score results on WolBanking77. LASER3+CNN (tuning) denotes LASER3+CNN with CNN hyperparameter tuning. Best models are highlighted in lightgray.

| Model | 5k samples | | | All samples | | |
|---|---|---|---|---|---|---|
| | Precision | Recall | F1 | Precision | Recall | F1 |
| LASER3+MLP | 0.56 | 0.56 | 0.55 | 0.43 | 0.42 | 0.42 |
| LASER3+CNN | 0.01 | 0.05 | 0.01 | 0.01 | 0.05 | 0.01 |
| LASER3+CNN (tuning) | 0.53 | 0.50 | 0.49 | 0.33 | 0.32 | 0.32 |

F1-score of 1%. To improve the LASER+CNN model, we tuned the CNN model hyperparameters using the Ray Tune library [Liaw et al., 2018]. The tuned hyperparameters include the size of the final internal representation before the classification layer, the learning rate (lr) and the batch size.We performed 10 tuning trials, and for each trial, the Ray Library randomly sampled a combination of parameters using the ASHAScheduler, which terminates poorly performing trials early. The best hyperparameters obtained are reported in Table 9. This leads to the improved results reported in Table 11.

## A.11   Consent form

Figure 6 presents the consent form (in French) presented to participants before the recording sessions.

**Formulaire de consentement**

Je, soussigné(e) X, déclare accepter librement, et de façon éclairée, de participer comme sujet à l'étude dans le cadre du
...........................................................................................................

Le linguiste, ....................................................., en charge de la collecte,
s'engage à protéger l'intégrité du participant tout au long de l'étude. Le participant donne l'autorisation au linguiste de partager les enregistrements avec la communauté des chercheurs ou des locuteurs de la langue pour des fins culturels et scientifiques. Le linguiste s'engage à ne pas diffuser des contenus potentiellement sensibles. Le participant peut décider à tout moment de restreindre l'accès aux enregistrements. Le consentement pour poursuivre la recherche peut être retiré à tout moment sans donner de raison et sans encourir aucune responsabilité ni conséquence. Les réponses aux questions ont un caractère facultatif et le défaut de réponse n'aura aucune conséquence pour le sujet. Le participant a la possibilité d'obtenir des informations supplémentaires concernant cette étude auprès du linguiste, et ce dans les limites des contraintes du plan d'étude. Toutes les informations concernant le participant sont conservées de façon anonyme et confidentielle. Le linguiste s'engage à préserver absolument la confidentialité pour toutes les informations concernant le participant.

Le participant : X                                                                 Le linguiste :

Fait le 24/05/2024 à :                                                         Fait le 24/05/2024 à :

Signature :                                                                         Signature :

Figure 6: Consent form

## B   Limitations

We can note some limitations in our dataset, such as class imbalance in intents, which may cause the model to favor majority classes at the expense of minority ones. During the ASR evaluation, we noticed spelling errors in rare words, which could be corrected using a language model.

## C   Ethics Statement

To protect the anonymity of participants, personal data such as speaker names, locations and androidIDs have been removed from the dataset. We notified each participant that the collected audio will be used as a public dataset for research purposes. Each participant read and signed a consent form in which they declared their free and informed consent to participate in the study as part of the data collection described in Section A.11. A total amount of $5000 was paid to the linguists for the translation and phonetic transcription of all sentences in Wolof and French.

## D   Wolof language

Wolof is a member of the West Atlantic sub-branch of the Niger-Congo language family. A small minority from various other ethnic groups have adopted Wolof as their first language, while more than half of non-native speakers use Wolof as their second or third language. Together, these three groups of Wolof speakers represent about 90% of the population, making Senegal one of the most linguistically unified nations in West Africa [Mbodj, 2014]. Wolof is also spoken by the majority of the population in Gambia [Ndione, 2013]. Genetically related languages are Pulaar and Serere, which are all languages of the Niger-Congo phylum, and of the Atlantic branch Ndione [2013] see figure 7.

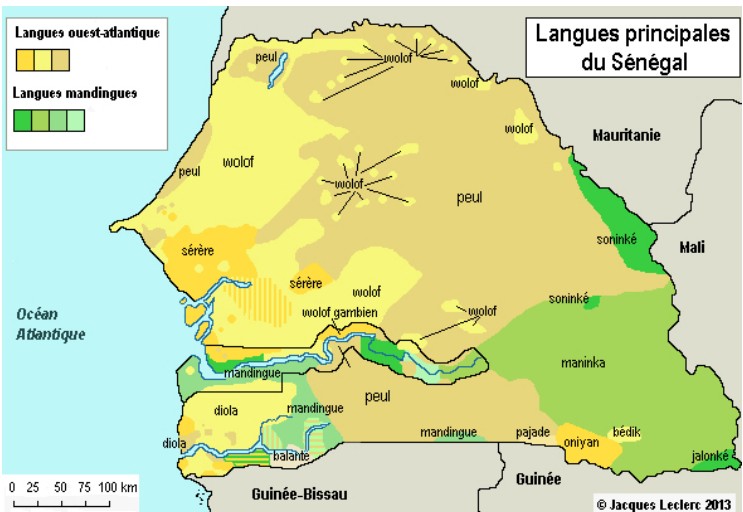

Figure 7: The main languages of Senegal Leclerc [2023]

To construct sentences in Wolof, there are several processes such as 8 class markers for words in singular form and 2 class markers for words in plural form. These markers are positioned in front of or right after the word such as: *nit ki* (to designate a person), *nit ñi* (to designate several persons). In addition, we also find other construction processes such as consonantal alternation, pre-nasalization and suffixation. For some derivations, there is the simultaneous presence of several of these processes. For example, pre-nasalization can be accompanied by suffixation [Ndione, 2013].

The conjugation in Wolof is achieved thanks to an invariable lexical base to which are added affixes holding the markers of IPAM (Index; Person Aspect-time; Mode) [Perrin, 2005, Ndione, 2013]. These inflectional markers highlight both semantic roles and grammatical relations. We mainly distinguish two types of verbs in Wolof, action verbs (expressing an action or a fact carried out or suffered by the subject) and state verbs (giving characteristics to elements, they describe a state, a way of being). There are ten conjugations in Wolof Perrin [2005], Ndione [2013] such as the perfect, the aorist, the presentative, the emphatic of the verb, the emphatic of the subject, the emphatic of the complement, the negative, the emphatic negative, the imperative and the obligative.

Depending on the grammar, in Wolof, there is generally only one auxiliary, "di", with its variants "d" and "y" which mark the unaccomplished. The variant "d" appears with the time and the negation markers (*doon, daa, daan, dee, du*).

## E   Datasheets for WolBanking77

### E.1   Motivation

**For what purpose was the dataset created? Was there a specific task in mind? Was there a specific gap that needed to be filled? Please provide a description.**

This dataset was created with the aim of propelling research on Intent Detection (ID) task in Wolof which is a language spoken in West Africa in order to allow illiterate people to be able to interact with digital systems through the voice channel. We introduce a text dataset and audio dataset including

utterances accompanied by their transcription with their corresponding intent in order to be able to simulate a dialogue between the user and the system.

**Who created the dataset (e.g., which team, research group) and on behalf of which entity (e.g., company, institution, organization)?**

This dataset is created by researchers at Université Cheikh Anta Diop of Dakar and Université Gaston Berger of Saint-Louis in Senegal.

**Who funded the creation of the dataset?**

Funding was provided by the Partnership for skills in Applied Sciences, Engineering and Technology (PASET) and Regional Scholarship and Innovation Fund (RSIF).

## E.2    Composition

**What do the instances that comprise the dataset represent (e.g., documents, photos, people, countries)? Are there multiple types of instances (e.g., movies, users, and ratings; people and interactions between them; nodes and edges)? Please provide a description.**

The dataset includes text and audio data for intent classification. For every sentence, text and annotations are provided.

**How many instances are there in total (of each type, if appropriate)?**

For the audio version, there are 263 instances covering 10 intents in total. 4 hours and 17 minutes of audios from spoken sentences. For the text version, there are 9,791 instances covering 77 intents in total.

**Does the dataset contain all possible instances or is it a sample (not necessarily random) of instances from a larger set? If the dataset is a sample, then what is the larger set? Is the sample representative of the larger set (e.g., geographic coverage)? If so, please describe how this representativeness was validated/verified. If it is not representative of the larger set, please describe why not (e.g., to cover a more diverse range of instances, because instances were withheld or unavailable).**

The dataset contain all possible instances.

**What data does each instance consist of? "Raw" data (e.g., unprocessed text or images)or features? In either case, please provide a description.**

Each instance consists of the text associated with intent label. The audio data contains transcriptions of the corresponding audio files.

**Is there a label or target associated with each instance? If so, please provide a description.**

Yes, there is a label intent class for each instance in the dataset.

**Is any information missing from individual instances? If so, please provide a description, explaining why this information is missing (e.g., because it was unavailable). This does not include intentionally removed information, but might include, e.g., redacted text.**

Everything is included. There is no missing data.

**Are relationships between individual instances made explicit (e.g., users' movie ratings, social network links)? If so, please describe how these relationships are made explicit.**

Each instance in WolBanking77 is relatively independent.

**Are there recommended data splits (e.g., training, development/validation,testing)? If so, please provide a description of these splits, explaining the rationale behind them.**

There are 80% of the data for training and 20% for validation. The intent categories were considered during the split by stratifying the data according to the intent target. This separation guarantees us to have all the intents both in the training set and in the test set.

**Are there any errors, sources of noise, or redundancies in the dataset? If so, please provide a description.**

The audio recordings were made in a controlled environment but may contain background noise.

**Is the dataset self-contained, or does it link to or otherwise rely on external resources (e.g., websites, tweets, other datasets)?**

WolBanking77 is self-contained.

**Does the dataset contain data that might be considered confidential (e.g., data that is protected by legal privilege or by doctor-patient confidentiality, data that includes the content of individuals' non-public communications)?If so, please provide a description.**

No.

**Does the dataset contain data that, if viewed directly, might be offensive, insulting, threatening, or might otherwise cause anxiety? If so, please describe why.**

No.

**Does the dataset identify any subpopulations (e.g., by age, gender)? If so, please describe how these subpopulations are identified and provide a description of their respective distributions within the dataset.**

The dataset identify subpopulations by age, gender and origin. The subpopulations are identified when participants fill the identification form by giving there age, gender and origin before starting a new recording session.

**Is it possible to identify individuals (i.e., one or more natural persons), either directly or indirectly (i.e., in combination with other data) from the dataset? If so, please describe how.**

No it is not possible to identify individuals from the dataset, names are removed from the dataset.

**Does the dataset contain data that might be considered sensitive in any way (e.g., data that reveals race or ethnic origins, sexual orientations, religious beliefs, political opinions or union memberships, or locations; financial or health data; biometric or genetic data; forms of government identification, such as social security numbers; criminal history)? If so, please provide a description.**

The dataset contain regions of origin showing that different voices and accents have been recorded to cover as more ethnic as possible and to make models robust to different accents.

### E.3  Collection Process

**How was the data associated with each instance acquired? Was the data directly observable (e.g., raw text, movie ratings), reported by subjects (e.g., survey responses), or indirectly inferred/derived from other data (e.g., part-of-speech tags, model-based guesses for age or language)? If the data was reported by subjects or indirectly inferred/derived from other data, was the data validated/verified? If so, please describe how.**

Dataset collection process has been reported in section 3.

**What mechanisms or procedures were used to collect the data (e.g., hardware apparatuses or sensors, manual human curation, software programs, software APIs)? How were these mechanisms or procedures validated?**

Data has been recorded using Lig-Aikuma Android software [Blachon et al., 2016].

**If the dataset is a sample from a larger set, what was the sampling strategy (e.g., deterministic, probabilistic with specific sampling probabilities)?**

Wolbanking77 is not sampled from a larger set.

**Who was involved in the data collection process (e.g., students, crowdworkers, contractors) and how were they compensated (e.g., how much were crowdworkers paid)?**

The data was collected with the help of students and professional translators. Data collection cost is around $5000.

**Over what timeframe was the data collected? Does this timeframe match the creation timeframe of the data associated with the instances (e.g., recent crawl of old news articles)? If not, please describe the timeframe in which the data associated with the instances was created.**

Our data collection started in March 2024. The contents of our dataset are independent of the time of collection.

**Were any ethical review processes conducted (e.g., by an institutional review board)? If so, please provide a description of these review processes, including the outcomes, as well as a link or other access point to any supporting documentation.**

Not applicable.

**Did you collect the data from the individuals in question directly, or obtain it via third parties or other sources (e.g., websites)?**

The text data is a translated version of Banking77 and the audio version from MINDS14.

**Were the individuals in question notified about the data collection? If so, please describe (or show with screenshots or other information) how notice was provided, and provide a link or other access point to, or otherwise reproduce, the exact language of the notification itself.**

A consent form shown in section A.11 has been provided to each individuals.

**Did the individuals in question consent to the collection and use of their data? If so, please describe (or show with screenshots or other information) how consent was requested and provided, and provide a link or other access point to, or otherwise reproduce, the exact language to which the individuals consented.**

Yes, see previous question.

**If consent was obtained, were the consenting individuals provided with a mechanism to revoke their consent in the future or for certain uses? If so, please provide a description, as well as a link or other access point to the mechanism (if appropriate).**

Yes, see A.11.

**Has an analysis of the potential impact of the dataset and its use on data subjects (e.g., a data protection impact analysis) been conducted? If so, please provide a description of this analysis, including the outcomes, as well as a link or other access point to any supporting documentation.**

No.

### E.4 Preprocessing/cleaning/labeling

**Was any preprocessing/cleaning/labeling of the data done (e.g., discretization or bucketing, tokenization, part-of-speech tagging, SIFT feature extraction, removal of instances, processing of missing values)? If so, please provide a description. If not, you may skip the remaining questions in this section.**

Dataset construction and cleaning has been reported in section 3. Text tokenization and removal of duplicated sentences has been done for NLP task.

**Was the "raw" data saved in addition to the preprocessed/cleaned/labeled data (e.g., to support unanticipated future uses)? If so, please provide a link or other access point to the "raw" data.**

No. The dataset does not contain all the raw data and metadata.

**Is the software used to preprocess/clean/label the instances available? If so, please provide a link or other access point.**

Python programming language has been used to clean the data. All source codes are available on Github.

### E.5 Uses

**Has the dataset been used for any tasks already? If so, please provide a description.**

Not yet.

**Is there a repository that links to any or all papers or systems that use the dataset? If so, please provide a link or other access point.**

No yet.

**What (other) tasks could the dataset be used for?**

The dataset can be used for anything related to intent classification as well as to train speech and text models.

**Is there anything about the composition of the dataset or the way it was collected and preprocessed/cleaned/labeled that might impact future uses? For example, is there anything that a dataset consumer might need to know to avoid uses that could result in unfair treatment of individuals or groups (e.g., stereotyping, quality of service issues) or other risks or harms (e.g., legal risks, financial harms)? If so, please provide a description. Is there anything a dataset consumer could do to mitigate these risks or harms?**

Not applicable.

**Are there tasks for which the dataset should not be used? If so, please provide a description.**

Not applicable.

## E.6    Distribution

**Will the dataset be distributed to third parties outside of the entity (e.g., company, institution, organization) on behalf of which the dataset was created?**

The dataset is distributed publicly on Kaggle.

**How will the dataset will be distributed (e.g., tarball on website, API, GitHub)? Does the dataset have a digital object identifier (DOI)?**

The dataset is distributed on Kaggle with DOI [KANDJI et al., 2025].

**When will the dataset be distributed?**

The dataset is published on Kaggle.

**Will the dataset be distributed under a copyright or other intellectual property (IP) license, and/or under applicable terms of use (ToU)? If so, please describe this license and/or ToU, and provide a link or other access point to, or otherwise reproduce, any relevant licensing terms or ToU, as well as any fees associated with these restrictions.**

WolBanking77 is distributed under the CC BY 4.0 [18] license. This license requires that reusers give credit to the creator. It allows reusers to distribute, remix, adapt, and build upon the material in any medium or format, even for commercial purposes.

**Have any third parties imposed IP-based or other restrictions on the data associated with the instances? If so, please describe these restrictions, and provide a link or other access point to, or otherwise reproduce, any relevant licensing terms, as well as any fees associated with these restrictions.**

No.

**Do any export controls or other regulatory restrictions apply to the dataset or to individual instances? If so, please describe these restrictions, and provide a link or other access point to, or otherwise reproduce, any supporting documentation.**

No.

## E.7    Maintenance

**Who is supporting/hosting/maintaining the dataset?**

---

[18]https://creativecommons.org/licenses/by/4.0/legalcode.en

Maintenance will be supported by the authors.

**How can the owner/curator/manager of the dataset be contacted (e.g., email address)?**

Authors can be contacted by their email address.

**Is there an erratum? If so, please provide a link or other access point.**

Any updates will be shared on Kaggle.

**Will the dataset be updated (e.g., to correct labeling errors, add new instances, delete instances)? If so, please describe how often, by whom, and how updates will be communicated to users (e.g.,mailing list,GitHub)?**

If necessary, any updates will be released on Kaggle.

**If the dataset relates to people, are there applicable limits on the retention of the data associated with the instances (e.g., were the individuals in question told that their data would be retained for a fixed period of time and then deleted)? If so, please describe these limits and explain how they will be enforced.**

N/A.

**Will older versions of the dataset continue to be supported/hosted/maintained? If so, please describe how. If not, please describe how its obsolescence will be communicated to dataset consumers.**

Older versions of the dataset will be kept.

**If others want to extend/augment/build on/contribute to the dataset, is there a mechanism for them to do so? If so, please provide a description. Will these contributions be validated/verified? If so, please describe how. If not, why not? Is there a process for communicating/distributing these contributions to dataset consumers? If so, please provide a description.**

Yes. Please contact the authors of this paper for building upon this dataset.

