# OpenReview forum: "WolBanking77: Wolof Banking Speech Intent Classification Dataset"
_NeurIPS.cc/2025/Datasets_and_Benchmarks_Track — NeurIPS 2025 Datasets and Benchmarks Track poster_

### Official Review · Reviewer_eWH8 · 2025-07-02

**Rating:** 4
**Confidence:** 5

**Summary:**

The paper is about intent classification for Wolof language, a language spoken in Senegal by about 90 million people. Given that Wolof is low-resource African language, it is underrepresented in NLP, LLMs and AI research. To this end, authors make a contribution towards increasing the representation of Wolof in the field by creating a new Wolof intent classification dataset Wolbanking77 consisting of 9,791 text sentences in the banking domain and more than 4 hours of spoken sentences to facilitate research in Wolof language.

**Additional Feedback:**

1. There are some grammatical errors that needs to be addressed to improve the paper.
 2. Also some terms like "illiterate" used in the abstract needs more clarification, how do you define illiteracy? The assumption that people are illiterate in regions where language is mostly spoken and not written is not true otherwise there is no reference or evidence in the paper supporting the use of the term "illiterate" i.e., in the African setting knowledge used and is still being passed on through oral communication.

**Dataset Code Accessibility:**

Yes

**Dataset Code Comments:**

The dataset is publicly available and can be readily accessed via Kaggle: https://www.kaggle.com/datasets/6f4251e190df4bb2c531856486d30b80c619155d2906f8fb3cd4448477a901b9.

**Ethical Considerations:**

No, there are no or only very minor ethics concerns

**Limitations Weaknesses:**

While the the authors conduct a wide range of experiments using encoder based models as well as Africa-centric models like AfroXLMR, AfroLM and AfritevaV2, it would be interesting and improve the paper if Africa-centric decoder models models like Lugha-Llama: https://arxiv.org/pdf/2504.06536, and AfriInstruct: https://aclanthology.org/2024.findings-emnlp.793.pdf are also evaluated.

**Strengths Contributions:**

The study makes releases a Wolof intent classification dataset, this is a significant contribution towards increasing the representation of Wolof language on NLP, LLMs and AI research. The authors also evaluate the performance of state-of-the art language models on the Wolobanking77.

The authors cite and explain the existing studies, they also make a good literature study of the existing and related studies and how they address the problem.

The paper is well written, with clear figures, and the experimental results support the findings. The authors also provide a detailed datasheet for Wolofbanking77.

---

> ### Author Rebuttal · Authors · 2025-07-31
>
> **We thank the reviewer for the interest in our work and the very valuable questions.**
>
> *__While the the authors conduct a wide range of experiments using encoder based models as well as Africa-centric models like AfroXLMR, AfroLM and AfritevaV2, it would be interesting and improve the paper if Africa-centric decoder models models like Lugha-Llama, and AfriInstruct.__*
>
> We train Lugha-Llama with LoRA using the same prompt as for Llama3.2 (Appendix A.3).
> Lugha-Llama is finetuned on WolBanking77 as a generative model for 3500 steps.
> Lugha-Llama then generate the label as the next token. This process reaches the following F1-score on 5k Split:
>
> F1 Score: 0.2470891046021977
>
> AfriInstruct is trained as a text classifier by adding a score layer for label intent classifier. Lora has been used to finetune AfriInstruct on WolBanking77. This process reaches the following F1-score on 5k Split:
>
> F1 Score: 0.0026200633618606453
>
> From the short experiments we could run it seems decoder-based models are less suitable for low-resource language classification. This should be investigated further to be confirmed.
>
> *__There are some grammatical errors that needs to be addressed to improve the paper.__*
>
> We will correct all the grammatical errors and request support for proofreading our article in order to have a high quality publication.
>
> *__Also some terms like "illiterate" used in the abstract needs more clarification, how do you define illiteracy? The assumption that people are illiterate in regions where language is mostly spoken and not written is not true otherwise there is no reference or evidence in the paper supporting the use of the term "illiterate" i.e., in the African setting knowledge used and is still being passed on through oral communication.__*
>
> We use the word “illiterate” in the sense of not being able to understand the official language, which is French. “ In Senegal, ANSD (2021) reports an overall illiteracy rate of 48,2%, reaching 62,7% in rural area. Literacy rate relates to the official language of a country. In Senegal, the official language is French but is seldom spoken by the population in their daily lives. Senegalese people primarily use their native languages or Wolof, as a vehicular language, to communicate.” Kallaama Dataset, Gauthier et al., 2024.
>
> The school dropout rate is high in Senegal and this has resulted in a significant number of people who went to school and left without acquiring basic skills in the official language (French). As a result, in our situation, basic skills in French are lacking for more than half of the Senegalese population. One of the solutions has been to resort to literacy in national languages with satisfactory results with the experiments in Pulaar carried out by NGO TOSTAN and the examples of functional literacy carried out by UNESCO. Despite this, a large segment of the population still remains in a situation where basic linguistic skills, either in the official language or in one of the national languages, are acquired very little or not at all.
>
> Literacy, considering the various languages present, but also the formal and informal systems, is progressing in Senegal even if it remains below 60% for adults according to statistics from UNESCO or the World Bank. According to these same sources, we note that among young people aged 15 to 24, the rate is higher and is around 78%, a sign of this progression. It is clear that there is a disparity on two levels, firstly, at the gender level, in fact adult women are far more victims of illiteracy than men of the same age group and also, urban areas suffer less than rural areas (UNESCO).
>
> Gauthier, E., Ndiaye, A., & Guissé, A. (2024, May). Kallaama: A Transcribed Speech Dataset about Agriculture in the Three Most Widely Spoken Languages in Senegal. In Proceedings of the Fifth Workshop on Resources for African Indigenous Languages@ LREC-COLING 2024 (pp. 10-19).

---

> > ### Comment · Reviewer_eWH8 · 2025-08-05
> >
> > I thank the authors for the feedback, I have read the comments from the authors and other reviewers and have decided to maintain my original rating.

---

### Official Review · Reviewer_gEzs · 2025-07-02

**Rating:** 5
**Confidence:** 5

**Summary:**

This paper introduces WolBanking77, the first intent classification dataset for Wolof, a West African language spoken by over 10 million people. The motivation is clear: 90% of Senegal's population speaks Wolof, but 42% are illiterate, creating barriers to digital banking services.

The dataset has two components: (1) A text dataset with 9,791 sentences covering 77 banking intents, translated from English Banking77 by linguistic experts; (2) An audio dataset with 263 sentences covering 10 intents, recorded from 31 speakers across different Senegalese regions using the Lig-Aikuma tool.

The authors benchmark various models from traditional ML to modern language models, testing both ASR (achieving 0.59% WER with Canary) and intent classification (achieving 57% F1 with AfroXLMR) tasks.

**Dataset Code Accessibility:**

Yes

**Ethical Considerations:**

No, there are no or only very minor ethics concerns

**Final Justification:**

After reviewing all rebuttals, I am satisfied with the responses and have decided to keep my ratings

**Limitations Weaknesses:**

* The audio dataset contains only 4 hours of speech with 263 samples across 10 intents. This is nowhere near sufficient for training robust ASR models or supporting real banking applications that would need hundreds of different scenarios.

* Data collection was restricted to university students aged 22-36, but the target users are illiterate populations who likely have very different language usage patterns. This fundamental mismatch undermines the practical applicability.

* Even the best intent classification model achieves only 57% F1 on the full dataset. For banking applications where accuracy is critical, this performance level is concerning. The impressive 0.59% ASR WER is achieved on a tiny test set and may not generalize.

* While positioned as enabling voice banking, the 10 audio intents cover only basic scenarios. Real banking systems require handling complex business logic, error recovery, and integration with existing infrastructure - none of which is addressed.

* Class Imbalance Issues: The authors acknowledge severe class imbalance (200 vs 24 examples) but don't adequately address how this affects model performance in practical deployment.

**Strengths Contributions:**

* The work addresses a real societal need. Enabling voice-based banking for illiterate populations in West Africa could genuinely improve financial inclusion and reduce fraud risks in informal economies.

* Filling a Critical Gap: This is the first Wolof banking dataset, contributing valuable resources to an severely under-resourced language used by 10 million people. The research community badly needs more work on African languages.

* The text translation involved professional linguists with proper localization (e.g., "ATM" → "GAB"). The audio data captures dialectal diversity across different Senegalese regions, which is important for model robustness.

* The authors tested a wide range of methods, including Africa-specific models like AfroXLMR and AfroLM, providing useful baselines for the community.

* Open and Reproducible: Released under CC BY 4.0 with complete datasheets and code, following good research practices.

---

> ### Author Rebuttal · Authors · 2025-07-31
>
> **We thank the reviewer for the interest in our work and the very valuable questions.**
>
> *__The audio dataset contains only 4 hours of speech with 263 samples across 10 intents. This is nowhere near sufficient for training robust ASR models or supporting real banking applications that would need hundreds of different scenarios.__*
>
> We agree on the limited size of the current audio dataset, but we aim at providing all possible data modalities we can to address the main goal as described in the last answer for Reviewer cDWP: The main goal for creating this dataset is to design a speech chatbot for Wolof, in order to equip illiterate people from Senegal with an AI tool allowing them to access banking services, as well as buying a transit pass from public transportation, paying bills, etc. This speech chatbot will be based on a Speech-to-Text module, a LLM, and a Text-to-Speech module, targeting mobile money applications with simple scenarios at first. We want also to point out that the ASR models used in this paper were pre-trained on very large data sets. If we take Canary as an example, it was pre-trained on 85K hours of speech data, including 31K hours from public data. The training data is rich and multilingual, which facilitates the generalization capacity of this model regardless of the size of the dataset. We of course consider only zero-shot or few-shot scenarios.
>
> Future work will address more complex scenarios.
>
> *__Data collection was restricted to university students aged 22-36, but the target users are illiterate populations who likely have very different language usage patterns. This fundamental mismatch undermines the practical applicability.__*
>
> It is true that the audio dataset was collected from a student population. However, we took care to select students from various geographical areas of the country as mentioned in the paper in Figure 3. This diversity allowed us to collect various accents as well as male and female voices, which is an important factor when building a robust ASR model.
>
> *__Even the best intent classification model achieves only 57% F1 on the full dataset. For banking applications where accuracy is critical, this performance level is concerning. The impressive 0.59% ASR WER is achieved on a tiny test set and may not generalize.__*
>
> The results on the intent classification were expectable since we have not fine-tuned the hyperparameters of the SOTA models on purpose to define baselines. Our aim was to expose the interest of our dataset for the community since it is definitely not covered by the existing datasets, otherwise we would have obtained far better F1-score from SOTA models. These F1-scores reveal the challenge we face when working with low-resource data since SOTA models, even though pre-trained on some Wolof text data, are not satisfying. Regarding the ASR result, the idea is also to investigate baselines in a low-resource context.
>
> *__While positioned as enabling voice banking, the 10 audio intents cover only basic scenarios. Real banking systems require handling complex business logic, error recovery, and integration with existing infrastructure - none of which is addressed.__*
>
> The idea is to build a baseline with 10 basic audio intents. While insufficient for production, a small dataset is useful for validating concepts, fine-tuning models, prototyping systems, and designing workflows. We plan to vocalize the text dataset covering 77 intents so that it can be integrated into more complex scenarios. As we write this, Mozilla's CommonVoice platform has just added the Wolof language. This could be a great opportunity for the community to contribute to the vocalization of our dataset.
>
> *__Class Imbalance Issues: The authors acknowledge severe class imbalance (200 vs 24 examples) but don't adequately address how this affects model performance in practical deployment.__*
>
> Class imbalance should indeed affect the model's performance in production. This can have a negative impact on classes that are underrepresented. Solutions could be to reduce the number of classes so that the classes can be rebalanced, or to explore data augmentation techniques, class weighting, and collecting more targeted data.

---

> > ### Comment · Reviewer_gEzs · 2025-08-04
> >
> > Thanks for your reply. I will keep my ratings.

---

### Official Review · Reviewer_ogFn · 2025-07-03

**Rating:** 4
**Confidence:** 3

**Summary:**

This paper presents WolBanking77, a version of Banking77 translated from English into French and Wolof. The paper presents initial scores for intent classification performance on WolBanking77. WolBanking77 also includes several hundred audio recordings to evaluate spoken language understanding. The paper also presents ASR results for this audio data.

**Additional Feedback:**

- Some of the references are arxiv preprints, but the papers have been published. In these cases, it is helpful to use the formal conference/journal citation rather than the arxiv citation. For example the ArBanking77 paper: https://aclanthology.org/2023.arabicnlp-1.22/ .
- The original Banking77 paper presents accuracy scores. Why not present accuracy scores for WolBanking77 too?

**Dataset Code Accessibility:**

Yes

**Ethical Considerations:**

No, there are no or only very minor ethics concerns

**Final Justification:**

updated

**Limitations Weaknesses:**

- W1)  Some models appear to be improperly trained. For instance, the CNN model in the appendix has very low scores, indicating that not enough hyperparameter tuning was done (Table 10).
-  W2) Banking77 (the original English version) has been shown to contain errors and ambiguous samples (https://aclanthology.org/2022.insights-1.19.pdf). Has anything been done to fix these errors or ambiguous cases in the proposed dataset? This is my main worry with this dataset; namely (a) Banking77 may have intent categories that are not meaningfully different than others, (b) contains label errors (up to 14% according to the listed paper), and (c) contains ambiguous samples like "HOW LONG TO TAKE THE TIME TO SOLVE" (see Table 1 of that paper).
- W3) Details of some of the models are unclear. For instance BERT is listed as a zero-shot model. What framework is used to make BERT a zero-shot model?

**Strengths Contributions:**

- S1) the paper presents data for two tasks, intent classification and ASR for an under-studied language in NLP. The Wolof language is used by millions of people and has several dialects. The choice of data application (i.e., personal finance and banking) is well motivated.
- S2) The paper presents baselines using relevant models (including LLMs), and shows there is room for improvement by future researchers.

---

> ### Author Rebuttal · Authors · 2025-07-31
>
> **We thank the reviewer for the interest in our work and the very valuable questions.**
>
> *__Some models appear to be improperly trained. For instance, the CNN model in the appendix has very low scores, indicating that not enough hyperparameter tuning was done (Table 10).__*
>
> Indeed, both for the CNN model and the MLP, hyperparameter tuning was insufficient. Note that these models were trained from scratch with hyperparameters from the literature. Following your suggestions, we have tuned the CNN model hyperparameters relying on the Ray library. The hyperparameters to be tuned are: the size of the last internal representation before the classification layer, the learning rate (lr) and the batch size. We have run the tuning on 10 different trials, and for each trial, Ray Library randomly sampled a combination of parameters using ASHAScheduler which terminates early the trials performing badly. The best hyperparameters obtained are:
>
> {‘last_non_classification_layer': 128, 'lr': 0.013364046097018405, 'batch_size': 2}
>
> This leads to an extension of the table 10 in Appendix:
>
>
> |                    |                    |          5k       |                    |                    |All Samples |                    |
> | ------------- |:-------------:| ------------- |:-------------:| ------------- |:-------------:|:-------------:|
> | Model        | Precision      | Recall         | F1               | Precision    | Recall.         | F1               |
> | LASER3+CNN (with hyperparameter tuning)        | 0.53      | 0.50   | 0.49    | 0.33        | 0.32      | 0.32      |
>
> *__Banking77 (the original English version) has been shown to contain errors and ambiguous samples. Has anything been done to fix these errors or ambiguous cases in the proposed dataset? This is my main worry with this dataset; namely (a) Banking77 may have intent categories that are not meaningfully different than others, (b) contains label errors (up to 14% according to the listed paper), and (c) contains ambiguous samples like "HOW LONG TO TAKE THE TIME TO SOLVE" (see Table 1 of that paper).__*
>
> Similar utterances were removed and assigned to the correct label when translating the dataset into Wolof. This is why we went from 10,003 utterances (from the original Banking77 train set) to 9,791 utterances.
>
> *__Details of some of the models are unclear. For instance BERT is listed as a zero-shot model. What framework is used to make BERT a zero-shot model?__*
>
> As we mentioned in the article, we consider the models listed in Section 5.1.1 for zero-shot classification. We use the pre-trained Masked Language Modeling (MLM) models (i.e. encoder models) on the datasets mentioned in this section.
>
> We then use  these models to embed each sentence and to project it onto the embedding of the intent classes for WolBanking dataset without additional training.
>
> To run BERT model in zero-shot mode, we used the huggingface framework with a pipeline configured as "zero-shot-classification". This pipeline can be used to classify sequences into any of the specified class names. In this case, the pipeline uses the pre-trained model to perform inference and retrieve the logits at the model output. A softmax function is then applied to the logits to generate the probabilities of each class.
>
> *__Some of the references are arxiv preprints, but the papers have been published. In these cases, it is helpful to use the formal conference/journal citation rather than the arxiv citation. For example the ArBanking77 paper.__*
>
> We will take your suggestions into account and update the paper with all the relevant references.
>
> *__The original Banking77 paper presents accuracy scores. Why not present accuracy scores for WolBanking77 too?__*
>
> In the context of class unbalanced, the accuracy score is not a reliable measure, which is why we considered more robust metrics such as precision, recall and f1-score. This is confirmed by the poor accuracy scores we got:
>
> | Model  | Accuracy  |
> | ------------- |:-------------:|
> | BertBase      | 55    |
> | AfroXLMR       | 57   |
> | AfroLM      | 57  |
> | mDeBERTa-v3 |    52         |
> | AfritevaV2 |       50      |

---

> > ### Author Response · Authors · 2025-08-07
> >
> > Please do not hesitate to let us know if our answers are not satisfying or if you have any further question, we will be glad to be able to answer them.

---

### Official Review · Reviewer_cDWP · 2025-07-23

**Rating:** 5
**Confidence:** 3

**Summary:**

This paper presents the WolBanking77 dataset, which is constructed primarily for scenarios involving low-resource languages and high illiteracy rates. The dataset contains 9,791 text sentences and 263 spoken sentences in the banking and transport domain, and has been applied to two tasks: Automatic Speech Recognition and Intent Detection. A detailed benchmark for the use of this dataset is also established. And the paper is clearly structured in its writing, and the research content holds significant application value.

**Additional Feedback:**

All my questions are written in "Limitations Weaknesses". If the authors can address my confusion, I will raise my score.

**Dataset Code Accessibility:**

Yes

**Dataset Code Comments:**

In the paper, a link to the dataset is provided, and the code is included in the Supplementary Material. Additionally, the experimental configurations are described in both the main text and the appendix of the paper.

**Ethical Considerations:**

No, there are no or only very minor ethics concerns

**Final Justification:**

The authors' response has addressed my confusion. I have decided to increase my score to the "accept" status.

**Limitations Weaknesses:**

Overall, I appreciate this research, but there are still some minor issues as follows:
1. What is the difference between this study and the research by Mastel et al. which created datasets through translation? Is there a bias between these translated datasets and real data?
2. The quality of the data is largely dependent on CLAD and Lig-Aikuma. Are there any evaluation metrics for data quality? For example, how to determine whether the translated data is of high quality?
3. What are the innovations of this paper? Is it because the Wolof dataset was chosen? Other low-resource languages presumably have similar issues. What are the highlights and innovations in this paper compared to the generation and processing of other low-resource language datasets?
4. Among the 77 text intents and 10 speech intents, is there any association between these intents? Has the possibility of multiple intents existing in one sample been considered?
5. How many ways are there to split the data? Were intent categories taken into account when splitting the data?
6. The paper lists some methods for low-resource language datasets but fails to conduct a detailed comparison, such as differences in data construction and benchmarks.
7. Why is the transport dataset included?  The contextual differences between the transportation field and the e-banking field seem significant.

**Strengths Contributions:**

This paper provides WolBanking77 dataset, which includes 9,791 text sentences and 263 spoken sentences. These samples are extracted and processed from the Banking77 and MINDS-14 datasets respectively. The effectiveness of the dataset is verified by applying it to Automatic Speech Recognition and Intent Detection tasks, and comparing the performance of different models. Meanwhile, a relatively complete benchmark is constructed. The strengths are as follows:
1. The paper is logically clear, highly readable, and rich in content.
2. The constructed WolBanking77 dataset has significant application value.
3. The benchmarks in the paper provide a relatively comprehensive evaluation of this dataset.

---

> ### Author Rebuttal · Authors · 2025-07-31
>
> **We thank the reviewer for the interest in our work and the very valuable questions.**
>
> *__What is the difference between this study and the research by Mastel et al. which created datasets through translation? Is there a bias between these translated datasets and real data?__*
>
> Wolof is not supported in Google Translate API, unlike the languages used in the work of Mastel et al. which are Kinyarwanda and Swahili. It is possible to use Google Translate web interface, but the Wolof translation quality is very poor. Wolof was integrated into Google Translate in 2024, and the results show that it is still in beta testing. Our objectives require high quality annotated data, this is why we opted for manual translation involving Linguistic experts.
>
> *__The quality of the data is largely dependent on CLAD and Lig-Aikuma. Are there any evaluation metrics for data quality? For example, how to determine whether the translated data is of high quality?__*
>
> The Centre de Linguistique Appliquée de Dakar (French for "Center of Applied Linguistics of Dakar"), abbreviated CLAD, is a language institute, which especially plays an important role in the orthographical standardization of the Wolof language. CLAD capitalizes on more than half a century of existence, marked by fundamental research in linguistics, the results of which are applied to language teaching and language planning with a view to contributing to facilitating the teaching and promotion of the languages spoken in Senegal. CLAD is composed of a team of researchers who are experts in translation into the Wolof language as well as other languages spoken in Senegal.
>
> *__What are the innovations of this paper? Is it because the Wolof dataset was chosen? Other low-resource languages presumably have similar issues. What are the highlights and innovations in this paper compared to the generation and processing of other low-resource language datasets?__*
>
> The WolBanking77 dataset, available in French and Wolof, is unique in the field of finance compared to other public datasets in African languages, which most often cover the fields of news or religion. Another important aspect is the manual data annotation, which was carried out by linguistic researchers, experts in Wolof. This method guarantees us the quality of data, which is crucial in the banking sector, instead of relying on automatic translation where the data quality is not guaranteed for low resource languages. Finally, intent detection is a subfield of natural language processing particularly underrepresented in low-resource languages even though it has a huge potential impact on real-case scenarios.
>
> *__Among the 77 text intents and 10 speech intents, is there any association between these intents? Has the possibility of multiple intents existing in one sample been considered?__*
>
> The text intent dataset and the speech intent dataset have been independently created. However, it could be possible to associate some speech intentions with text intentions such as BALANCE, CASH_DEPOSIT, FREEZE and TRANSFER_MONEY. It is indeed possible that a sample can belong to several intentions, however this possibility has not been considered in this work.
>
> *__How many ways are there to split the data? Were intent categories taken into account when splitting the data?__*
>
> Since the evaluation of the quality of our dataset is based on performances of state-of-the-art existing methods, we have not changed or fine-tuned the hyperparameters of these methods. Thus, we have not considered any validation set and split the data into a train set of 80% and a test set of 20%. The intent categories were considered during the split by stratifying the data according to the intent target. This separation guarantees us to have all the intents both in the training set and in the test set.
>
> *__The paper lists some methods for low-resource language datasets but fails to conduct a detailed comparison, such as differences in data construction and benchmarks.__*
>
> Please find below a detailed description of the differences in data constructions and benchmarks for the main datasets we consider in our paper:
>
> *__ALFFA__*
>
> Task: ASR
>
> Data construction: Audio recording of 21 hours including 18, 000 utterances and using a Samson G-track microphone in a clean environment
>
> Benchmark: 27.21 WER
>
> *__Kallaama__*
>
> Task: ASR, NLU, NLG
>
> Data construction: manual transcription of audio recordings comprising interactive radio programs, focus groups, voice messages, push messages and interviews. Transcriber was used as a tool for transcription. 55h of Wolof.
>
> Benchmark: no proper benchmark in the original paper.
>
> *__INJONGO__*
>
> Task: Intent Detection & Slot-filling
>
> Data construction: Manual utterance generation belonging to the same intent type on each language. INJONGO is inspired on CLINIC dataset, 40 intents across 5 domains (Banking, Kitchen and Dining, Travel, Utility, and Home) are extracted from CLINIC English dataset. 3,200 utterances with a balanced number of intent types has been collected for Wolof.
>
> Benchmark: Among the different models evaluated in the original paper, AfroXLMR-76L achieves the best performance on Wolof with a score of 97.1 on Intent classification.
>
> Our dataset contains more utterances in Wolof and cover more intent than INJONGO. Our dataset can been combined with INJOGO in order to enrich intent detection task in Wolof language in the banking domain. This domain is almost invisible because  the majority of the datasets covers religion or news. Our data also covers speech, which is a modality not taken into account in INJOGO's work. Speech is an essential modality in African languages.
>
> *__Why is the transport dataset included? The contextual differences between the transportation field and the e-banking field seem significant.__*
>
> The main goal for creating this dataset is to design a speech chatbot for Wolof, in order to equip illiterate people from Senegal with an AI tool allowing them to access banking services, as well as buying a transit pass from public transportation, etc.
>
> Adding transportation data, is thus not only a way to evaluate the model's ability to recognize out of context intents, but also to integrate complementary services into a mobile money application. Beyond making financial transactions with mobile money, it is possible to integrate other services such as transportation or bill payments (electricity, water, etc.). The fields are different but the kind of services (and so the sentences) we want to address are pretty similar in all of them.

---

> > ### Author Response · Authors · 2025-08-07
> >
> > Please do not hesitate to let us know if our answers are not satisfying or if you have any further question, we will be glad to be able to answer them.

---

> > ### Comment · Reviewer_cDWP · 2025-08-07
> >
> > Thank you for your reply, which has resolved my confusion to a certain extent. I have decided to increase my score to the "accept" status.

---

### Decision · Program_Chairs · 2025-09-18

**Decision:**

Accept (poster)

**Comment:**

The paper introduces a small dataset in Wolof focusing on intent classification and speech recognition in the banking sector. It is a valuable contribution for an under-resourced language. It is especially notable that the dataset includes audio, as Wolof is primarily a spoken language.

There are some concerns, but they are minor and expected for new datasets in low resource languages. In particular, concerns focus on the fact that the dataset is small, that the modeling results are not very strong, and that there is more work needed to bridge the gap between this dataset and realistic applications. It would be nice to see such concerns addressed, but is understandable if that happens in future work.

All reviewers recommend accepting, and I agree.